# An Introduction to Assessing Dental Fear and Anxiety in Children

**DOI:** 10.3390/healthcare8020086

**Published:** 2020-04-04

**Authors:** Madeline Jun Yu Yon, Kitty Jieyi Chen, Sherry Shiqian Gao, Duangporn Duangthip, Edward Chin Man Lo, Chun Hung Chu

**Affiliations:** Faculty of Dentistry, The University of Hong Kong, Hong Kong; yonjunyu@hku.hk (M.J.Y.Y.); kjchen@hku.hk (K.J.C.); sherryg@hku.hk (S.S.G.); dduang@hku.hk (D.D.); hrdplcm@hku.hk (E.C.M.L.)

**Keywords:** dental fear, dental anxiety, children, assessment

## Abstract

Fear and anxiety constitute an important theme in dentistry, especially with children. Anxiety and the fear of pain during dental treatment can lead to avoidance behaviour, which contributes to perpetuating fear and anxiety of dental care. Understanding and assessing dental fear and anxiety in children is important for delivering successful dental care with high satisfaction in this age group. Among the vast assessment method options available today, self-report assessment, parental proxy assessment, observation-based assessment, and physiological assessment are the four major types for dental fear and anxiety in children. Each method has its own merits and limitations. The selection of a method should be based on the objectives, validity, and setting of the assessment. The aim of this paper is to review and discuss the assessment methods for dental fear and anxiety in children.

## 1. Introduction

Fear and anxiety constitute an important theme in dentistry, especially with children. Fear and anxiety are some usual reactions to stressful conditions. They may help children in staying alert in situations of impending threat. However, highly fearful or anxious children generally experience dental visits and treatments in a more negative way. They may experience various levels of apprehension before, during, and after dental treatment which may be mild and temporary or severe and affect an individual long before the date of the scheduled appointment. It has been suggested that such anxiety results in a vicious cycle whereby anxiety and the fear of pain during dental treatment leads to avoidance behaviour, which contributes to perpetuating fear and anxiety of dental treatment [1].

Several assessment tools have been introduced to investigate children’s psychological or behavioural changes due to dental fear and anxiety. Before going further into the discussion, it is essential to recognise that terminologies such as ‘fear’, ‘anxiety’, and ‘phobia’ have related and overlapping connotations yet are not entirely the same. According to the Cambridge Dictionary, both ‘fear’ and ‘anxiety’ are described as being ‘unpleasant’ and related to ‘worry’ [2]; comparison of the two terminologies were not deliberately made. In the dental literature, however, an attempt has been made to specify the meaning of each term. For example, fear was regarded as a reaction and a response to an immediate live threat. It can be mitigated by reassurance, or it can be reduced to a certain degree by training [3]. Fear is a psychological construct which is not directly observable but may manifest itself through cognitive, behavioural and physiological responses according to the cognitive vulnerability model of fear [4]. For example, the sight of a needle or the sound of drilling can cause the feeling of fear. This feeling stems from specific stimuli, in contrast to anxiety. Anxiety is understood as a feeling of fear or apprehension about what is to come. Anxiety arises internally, and the cause of it may not be immediately apparent from the surrounding environment [5]. Anxiety such as that in dental fear and anxiety is an internal state which is not directly accessible to objective ways of measurement [6]. 

Phobia, on the other hand, is a mental disorder diagnosis and is diagnosed when a person persistently exhibits marked or extreme fear or anxiety responses that might be out of proportion, or cause “marked distress” [7]. It is a heightened fear and anxiety that is serious enough to disrupt a person’s normal routine, such as leading to the avoidance of dental treatment [8,9]. Such an irrational and excessive fear reaction and is often connected to something specific, which can be of a certain place, situation or object. Children with phobias will experience a deep sense of dread or panic when they encounter the sources of their fear. Due to the similarity between terminologies and their mixed usage in the English language, these concepts have often been discussed together or under the umbrella term of dental fear and anxiety [10,11].

Understanding dental fear and anxiety in young children is an essential part of reducing their fear and anxiety pre- and peri-operatively, as well as in managing child patient behaviour. Children who are relaxed and unafraid may display positive behaviour, such as smiling and chatting. On the contrary, fearful children may behave hesitantly or cautiously, while some may defend themselves physically and disrupt treatment procedures in extreme cases. Appropriate assessment methods are crucial to understanding different aspects of dental fear and anxiety in children. The aim of this paper is to review and discuss assessment methods for dental fear and anxiety in children.

## 2. Assessment Methods for Dental Fear and Anxiety in Children 

The assessment of dental fear and anxiety in children is possible through a number of means. The determination of the presence or severity of dental fear and anxiety in children requires more indirect methods, such as asking the child questions (cognitive approach), observing behaviour during dental treatment, or recording physiological responses to anxiety, such as the pulse rate and sweating. Amongst the vast number of options, four major types of assessment tools may be grouped based on the type of informant or information gathered [12]. They are self-report assessment, parental proxy assessment, observation-based assessment, and physiological assessment.

### 2.1. Self-Report Assessment for Dental Fear and Anxiety

The most common dental fear and anxiety assessment measures belong to the self-report type. Children respond to questions or instructions that provide information about their dental fear and anxiety. No other parties, such as the children’s parents, are involved. The simplest way is to directly ask a single question: whether the child is scared or anxious, of course. Alternatively, a visual analogue scale may assist in pointing out the severity of dental fear and anxiety. However, more sophisticated means of dental fear and anxiety measurement have been used in research. This may be in the form of questionnaires, pictorial scales or the assessment of children’s artwork in a guided situation.

The Dental Subscale of the Children’s Fear Survey Schedule is a psychometric scale questionnaire used to assess dental fear in children using fifteen questions. The Modified Dental Anxiety Scale is a brief, self-completed questionnaire used to screen for the dental anxiety of patients using five questions. These questionnaires commonly consist of scenario-type questions [12,13]. They can present certain levels of difficulty and not suitable for of still-developing children.

Pictorial scales have been developed to overcome this problem, especially in younger children [14], thus allowing a child to respond non-verbally [6]. The Venham Picture Test is a projective, psychometric, self-measure test which is used to measure the level of dental anxiety of a young child. The pictorial scale consists of eight pairs of pictures, each containing two cartoon figures of a boy with boldly drawn facial expressions depicting different emotions [6]. Another scale, the Children’s Dental Fear Picture Test, contains a section of cartoons with contrasting emotions. Even simpler still is the Facial Image Scale, which consists of a range of five faces with facial expressions ranging from very unhappy to very happy [14]. A child simply needs to point to the cartoon that best represents himself or herself at that moment at the instructor’s cue [15]. The assessment tools listed above are close ended in nature, providing a framework or a list of questions for the child to answer. 

An open-ended assessment of dental fear and anxiety involving children’s drawings has been suggested. Children’s drawings may assist the child in elaborating their feelings where words fail [16]. A study suggested the analysis of these drawings as a method for assessing the dental fear and anxiety of young children [5]. Children were given drawing materials and some time to draw a dental clinic after finishing their dental treatment. Their drawings were analysed according to a scoring sheet with criteria such as how humans were drawn and whether the exaggeration or omission of an object in the dental clinic was present. The final score reflected the level of dental anxiety that the child felt [5].

### 2.2. Parental Proxy-Based Assessment for Dental Fear and Anxiety

Specific to the paediatric assessment of dental fear and anxiety is the category of parental proxy-type measures. As the name suggests, respondents are the parents of children whose dental fear and anxiety investigators intend to measure. The use of parental proxy is intended to bypass the difficulty of comprehension or the lack of cognitive ability in children [8], especially younger children [17]. Parents can be invited to describe their children’s anxiety from their perspectives based on their past impressions and experiences [18,19,20]. A frequently used measure is the parent version of the Dental Subscale of the Children’s Fear Survey Schedule, where the wordings were changed to “How afraid is your child of [different scenarios in a dental clinic]?” When parental proxy-based assessment tools are used, the bias of using parents’ perceptions of their children’s dental fear and anxiety should be borne in mind and considered during analysis [21]. This is because the correlation between parents’ responses and their children’s dental fear and anxiety has been shown to vary greatly depending on the children’s developmental stages [22]. 

### 2.3. Observation-Based Assessment for Dental Fear and Anxiety

Another method of dental fear and anxiety assessment is to observe the child throughout dental treatment and to assess a child’s dental fear and anxiety according to his or her behaviour or facial expressions, usually relying on a descriptive scale. The child is not required to answer specific questions about his or her dental fear, and the observer may be a person unrelated to the treatment or to the dentist himself or herself [23,24,25]. Some examples of observation-based dental fear and anxiety assessment tools are the Clinical Anxiety Rating Scale/Uncooperative Behaviour Rating Scale (CARS/BRS) [26], Frankl Behaviour Rating Scale (FBRS) [15,25,27], Behavior Evaluation Scale (BES) [28], Behavior Profile Rating Scale (BPRS) [29], Houpt scale [30], and Verbal Skill Scale (VSS) [31]. The CARS/BRS and FBRS are descriptive scales to classify child’s fear or anxiety at a few levels based on behaviour at the dentist’s office. The rating is given through generalising the child’s behaviour into ‘positive’ vs. ‘negative’ (FBRS) or multiple levels (0–5, CARS/BRS). Another scheme, such as that presented in BES, is to give a score for every action or expression from the child during dental treatment from a checklist, such as marking down ‘yes’ when the child rolled his eyes or shook his head. It may be more difficult to administer such tests in practice due to the long list of items, and it might be challenging to observe every action on the checklist [32]. The BPRS is even more comprehensive as score calculation involves multiplication of items on the checklist with individual weightings, which reflects their level of disruptiveness, to give an aggregate score [33]. Though the selective items act as a guide, full score calculation may be non-feasible in non-research settings. Grouping behaviour in four categories and adding subscores in each category, such as that in the Houpt scale, simplifies the assessment procedure and improves practicality. The VSS approaches dental fear differently by observing the level and type of child’s eye contact, verbal response, and facial expression and marking down on a score sheet during the child’s conversation with the dental operator. To reduce the bias of observation and scoring, training and calibration are usually recommended, and observer bias may be minimised by videotaping and scoring outside of the treatment session [31,34]. 

### 2.4. Physiological Assessment for Dental Fear and Anxiety

Researchers also assess dental fear and anxiety through the direct measurement of the physiological status of the child. Certain psychological stress markers known to correlate with fear or anxiety levels could reflect the level of dental fear and anxiety. Various measures, such as pulse/heart rate, nervous activity, and muscular activity, may permit real-time and continuous measurement at different phases of treatment [15,35,36]. Others, such as levels of psychological stress markers in saliva and serum or palmar sweating, may be detected at specific time points during dental treatment [37,38,39]. Such methods do not require the child or the parent to respond to questions, but the investigator might need to attach monitoring equipment to the child subject who is undergoing dental treatment, with the process possibly inducing fear [40].

## 3. Selection of Assessment Method

In the previous section, a variety of assessment methods were given which provide multiple possibilities for understanding children’s dental fear and anxiety. Table 1 summarises the main advantages and limitations of dental fear and anxiety assessment tools for children. The selection of dental fear and anxiety assessment tools for children should be made based on the objectives, age of study target, validity, and the way that the assessment is carried out.

### 3.1. Objectives of the Assessment

The aim of the investigation or intended outcome should be the main driver in our decision as to which dental fear and anxiety measure to use for children. If the researcher’s interest is in children’s intrinsic fear of dentistry in general, the instrument chosen should tap into this internal state of anxiety to understand the psychology of dental fear. Where the purpose is to determine children’s anxiety response and behaviour to various dental-related stimuli for facilitating treatment planning, dental cues of fear may become the focus of the investigation, and question sets concerning behaviour or behaviour rating scales in a well-defined dental situation should be considered [38], and the chosen scale might tend to be more operator-centred (e.g., how the child’s fear and anxiety disrupted the dental procedure) rather than patient-centred (e.g., how the child felt and thought during the dental procedure) [41]. Additionally, open questions regarding the child’s past experiences in specific dental procedures might be practically useful for the clinician.

If it is the researcher’s intent to conduct an epidemiological study on the topic of children’s dental fear, reliable and valid instruments with well-defined cut-off scores should be utilised, which is just as important as the stringent choice of sample population and condition.

### 3.2. Age of Study Target

The age of the study target is related to the objective of dental fear assessment in children. As Locker has pointed out, the onset of dental anxiety could be divided by the age of an individual [42]. Needless to say, the choice of method inappropriate to the child subject’s age and level of development could affect the ability of the assessment tool to reflect the level of dental fear and anxiety accurately. Many studies have pointed out plainly that very young children could not reliably report aspects of their own dental fear due to a minimum cognitive ability required to understand questions and to respond meaningfully [43] and require assistance from parents or other adults in answering questions [44]. Another study indicated that assessment tools using abstract numeric scales pose another challenge to respondents younger than the age of nine who might fare better with pictorial scales [45]. Yet another concern with the respondent’s age is past dental experience. While adults may speak of their dental fear and anxiety from their own past experience(s), very young children often have no prior dental experience to refer to, including both fear- and non-fear-provoking situations, which invalidates evaluation methods based on past dental experience for this age group [22,46]. These factors should be taken into consideration when planning a study involving very young children. 

### 3.3. Validity of the Assessment

Many studies have been conducted on the translation and validity assessment of dental fear and anxiety assessment tools for children [47,48,49,50], yet the validity of several self-report questionnaires and pictorial scale remains questionable. This has been attributed to the lack of strong conceptual or theoretical backing [10,11], or to the improper choice of questions for young children. Items such as scraping and polishing are specific dental procedures that young children may not have experienced in the past, and others, such as “having to go to the hospital” in the Dental Subscale of the Children’s Fear Survey Schedule, may not be applicable to most dental situations [51]. Moreover, in recording the fear or anxiety responses of children, a number of scales (e.g., Venham Picture Test, Clinical Anxiety Rating Scale/Uncooperative Behaviour Rating Scale) could describe the severity of dental fear and anxiety, but each has only a single score for positive experience. Meanwhile, the Facial Image Scale and the Frankl Scale are able to capture the degree of positive feeling or experience by letting the child choose a ‘happy’ or ‘very happy’ face, or the observer may rate a child’s behaviour as ‘positive’ or ‘definitely positive’ [52,53]. 

Although questionnaires and text-based self-report measures could be limited by language, pictorial scales might present other ambiguities. For example, the Venham Picture Test was developed with only male cartoons and exaggerated facial expressions. The drawing style might also affect children’s responses despite the attempt to avoid identifiable racial or socioeconomical characteristics [6]. The use of facial images that are detached from drawings showing clinical environments or treatment procedures and showing only ‘happy’ or ‘unhappy’ emotions might avoid this problem. 

Other studies have attempted to show concurrent validity between dental fear and anxiety measures [31,54]. It is, however, normal to expect such a correlation not to be perfect, especially among instruments that measure different components of dental fear and anxiety. These include those focused on internal fears (cognitive) versus instruments measuring outwardly visible behaviour and physiological responses, as well as measures focused on trait anxiety whereas others measuring state anxiety, which are fundamentally different. This, again, highlights the multidimensionality of children’s dental fear and anxiety, as well as the need for careful selection (or combination) prior to conducting a study on children’s dental fear and anxiety [38].

### 3.4. How the Assessment is Conducted

Apart from the design and properties of each individual dental fear and anxiety assessment tool, the way in which the assessment is conducted must be considered when planning an investigation. 

Past studies involved conducting dental fear and anxiety assessments of children in various settings, such as dental clinics and specialist referral centres [24,34,40], or in non-clinical environments, such as homes and schools [18,21]. The sample population included the general population [18,21,47], children in general dentistry [15,24], and those referred for specialist care, including those referred based on dental behavioural and management problems [12,34,40]. The severity and prevalence of dental fear and anxiety must vary among these populations, and different cut-off points and categorisation should be adopted [12]. Other than study settings, details of the dental fear and anxiety assessment procedures should be foreseen and planned. Do the study subjects fill out questionnaires independently, or are they helped by parents at home who might impress upon them their own perceptions of dental anxiety [21]?

How procedure is conducted also impacts on the reliability of dental fear and anxiety assessment. For example, consider the people administering the test, performing the scoring, or simply being present at the scene of the dental fear and anxiety assessment, be they assistants or the dentist himself or herself. It has been described in literature that children might respond differently to dentists who are less receptive, such as giving a ‘frozen response’ to them rather than behaving in an agitated manner [40]. This illustrates that controlling for the observer, treatment provider and treatment protocol, including but not limited to spoken instructions, becomes essential in maintaining the reliability of the measure. It is advisable to arrange training and calibration for multiple observers or even to have the same observer throughout to improve reliability. Additionally, video-recording the process for repeated scorings is a well-documented and recommended strategy when proper consent has been obtained [24,34].

## 4. Conclusions

In conducting a proper assessment of dental fear, the selection of the assessment method and matching the measurement method to the aim of the study are only the first steps. It is vital to also appreciate the characteristics of existing methods for dental fear and anxiety assessment and to avoid pitfalls through rigorous study planning. Notwithstanding the challenges presented in this paper, it is possible to appraise and gain new knowledge of children’s fear and anxiety in dentistry when assessments are conducted properly.

## Figures and Tables

**Table 1 healthcare-08-00086-t001:** Advantages and limitations of the methods for assessing children’s dental fear and anxiety.

Types (Examples)	Advantages	Limitations
Self-report assessment (CFSS-DS, MDAS, CDFPT, drawings)	Measures cognitive element component of DF directly from child’s perspective Drawing activity may allow children to elaborate their perspectives of DFA Can be used in non-clinical settings, e.g., in school or at home	Developmental maturity may affect the ability for self-reporting and reliability in reporting. Adequate level of comprehension and intellectual ability required Interpretation of children’s drawings may require professional training.
Parental proxy–based assessment (Parent version of CFSS-DS)	Presents adult’s perspective of child’s DFA Parents may have knowledge of child’s concerns and past behaviour.	May form bias through their own DFA and dental experiences
Observation-based assessment (CARS/BRS, FBRS, BES, BPRS, Houpt scale, VSS)	Measures behavioural component of fear Presents adult’s perspective of child’s DFA More objective when scoring criteria could be followed Repeated measure and scoring may be possible, e.g., by means of videotaping.	Describes only observable signs of DFA Training or calibration, especially when multiple observers are involved, to minimise observer effect or bias
Physiological assessment (Stress markers, such as heart rate)	Measures physiological component of DFA Most objective measure of DFA	Requires specialised equipment Must be performed during the dental visit (instead of in non-clinical settings) Observer effect due to invasive and fear-provoking measuring procedures

DFA: Dental fear and anxiety; CFSS-DS: Dental Subscale of the Children’s Fear Survey Schedule; MDAS: Modified Dental Anxiety Scale; CDFPT: Children’s Dental Fear Picture Test; CARS/BRS: Clinical Anxiety Rating Scale and Uncooperative Behaviour Rating Scale; FBRS: Frankl Behaviour Rating Scale; BES: Behavior Evaluation Scale; BPRS: Behavior Profile Rating Scale; VSS: Verbal Skill Scale.

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
