# Peer review of "An Introduction to Assessing Dental Fear and Anxiety in Children"

_healthcare, 2020, doi:10.3390/healthcare8020086_

Round 1

Reviewer 1 Report

The Study is good and interesting. The aim of this review article is to analyse different evaluation scales that are most commonly used to assess dental anxiety and fear in children. This systematic review identified the psychometric properties and feasibility of the principal types of these assessment methods in children. There is no universally recommended method that could be used in all contexts for anxiety and fear evaluation during treatment among pediatric patients.The manuscript can be clearer if explained elaborately. 

  1. I think it would be good to extent and point out more assessment tools, developed and used nowadays, for example - Kurosu Behaviour Evaluation Scale (in the group of the observation-based methods). Here are several articles that you come familiar with it - Shinohara S,Nomura Y, Shingyouchi K, et al. Structural relationship of child behavior and its evaluation during dental treatment. J Oral Sci 2005 Jun; 47(2):91-6., Shinohara S, Nomura Y, Ide M, et al. The classification of the children by their behavior for the dental treatment using cluster analysis. J Pediatr Dent 2005; 15:191-194.
  2. There is one more very important factor in the selection of an assessment method for children - the age! It is necessary to mention it and describe how to classify self-assessment tools according to age of the investigated group. It is necessary to introduce some scale that are modified according to children`s age (for example - Shindova M., Belcheva A., Mateva N. Factors in dental environment related to development of child dental fear and parent-child agreement on its evaluation. International scientific on-line Journal Science & Technologies. Medical biology studies, clinical studies, social medicine and health care. Volume IV, 2014,91-95.).
  3. Please add recent literature

Author Response

Comments: The Study is good and interesting. The aim of this review article is to analyse different evaluation scales that are most commonly used to assess dental anxiety and fear in children. This systematic review identified the psychometric properties and feasibility of the principal types of these assessment methods in children. There is no universally recommended method that could be used in all contexts for anxiety and fear evaluation during treatment among pediatric patients.The manuscript can be clearer if explained elaborately. 

Response: Thank you for your valuable comments.

Comments: 1. I think it would be good to extent and point out more assessment tools, developed and used nowadays, for example - Kurosu Behaviour Evaluation Scale (in the group of the observation-based methods). Here are several articles that you come familiar with it - Shinohara S,Nomura Y, Shingyouchi K, et al. Structural relationship of child behavior and its evaluation during dental treatment. J Oral Sci 2005 Jun; 47(2):91-6., Shinohara S, Nomura Y, Ide M, et al. The classification of the children by their behavior for the dental treatment using cluster analysis. J Pediatr Dent 2005; 15:191-194.

Response: Agree. The relevant examples including the study of Shinohara et al.2005 are added to the manuscript in Page 3, Line 125-138.

Comments: 2. There is one more very important factor in the selection of an assessment method for children - the age! It is necessary to mention it and describe how to classify self-assessment tools according to age of the investigated group. It is necessary to introduce some scale that are modified according to children`s age (for example - Shindova M., Belcheva A., Mateva N. Factors in dental environment related to development of child dental fear and parent-child agreement on its evaluation. International scientific on-line Journal Science & Technologies. Medical biology studies, clinical studies, social medicine and health care. Volume IV, 2014,91-95.).

Response: Done. “Age” is included and elaborated in section 3.2 of the manuscript in Page 5 , Line 180-195.

Comments: 3. Please add recent literature
Response: Done. Twenty-five of the cited literature are from 2010 to now.

Reviewer 2 Report

This is a review of possible instruments used to assess Dental Fear and/or Anxiety, and their appropriateness and limitations in designing a relevant study. Some of them are simply mentioned, while some other are critically and briefly described. Some more are not mentioned, e.g. since Frankl scale is included, the BPRS (Behavior Profile Rating Scale) or the HBRS (Houpt Behavior Rating Scale) could be, too.  This is about the methodology of literature search that is not explained. There is a selection of references, done in a non-systematic way, serving this narrative review’s purpose. There are also some, not many, syntax mistakes.  

Title ‘Assessing dental fear and anxiety in children’ is too general and does not fully reflect the content of the text. As it is, it prepares the reader for a research paper. Perhaps adding: ‘’An introduction to assessment methods for…’’ would define it better.

There is an introduction to general fear, anxiety and phobia terms. The Cambridge Dictionary mentioned as not clearly distinguishing between fear and anxiety, does in fact offer a difference for fear (‘’… by something dangerous, painful, or bad’’) vs. anxiety.

The manuscript is quite helpful as a guide (this is also reflected in the conclusions) to a reader who wants to be introduced in assessing dental fear and anxiety in children. So is the table, though comments on children age and a more systematic lay out including the reliability and validity of each scale separately would increase the paper usefulness.

Author Response

Comments: This is a review of possible instruments used to assess Dental Fear and/or Anxiety, and their appropriateness and limitations in designing a relevant study. Some of them are simply mentioned, while some other are critically and briefly described. Some more are not mentioned, e.g. since Frankl scale is included, the BPRS (Behavior Profile Rating Scale) or the HBRS (Houpt Behavior Rating Scale) could be, too.  This is about the methodology of literature search that is not explained. There is a selection of references, done in a non-systematic way, serving this narrative review’s purpose. There are also some, not many, syntax mistakes.  

Response: Thank you for your valuable comments. The relevant section has been elaborated with inclusion of the listed examples including BPRS (Behavior Profile Rating Scale) or the HBRS (Houpt Behavior Rating Scale) in Page 5, Line 125.

Comments: Title ‘Assessing dental fear and anxiety in children’ is too general and does not fully reflect the content of the text. As it is, it prepares the reader for a research paper. Perhaps adding: ‘An introduction to assessment methods for…’ would define it better.

Response: Agree. The title has been revised following your suggestion.

Comments: There is an introduction to general fear, anxiety and phobia terms. The Cambridge Dictionary mentioned as not clearly distinguishing between fear and anxiety, does in fact offer a difference for fear (‘… by something dangerous, painful, or bad’) vs. anxiety.

Response: Agree. The introduction part has been modified to better reflect this in Page 1, Line 36-37.

Comments: The manuscript is quite helpful as a guide (this is also reflected in the conclusions) to a reader who wants to be introduced in assessing dental fear and anxiety in children. So is the table, though comments on children age and a more systematic lay out including the reliability and validity of each scale separately would increase the paper usefulness

Response: Revised. A new section on children’s age affecting the choice of assessment tool is added in Page 3. The layout and content have been revised in Page 3, 5 and Table1.